# Identification of Key Genes and Regulatory Pathways in Multiple Sclerosis Brain Samples: A Meta-Analysis of Micro-Array Datasets

**DOI:** 10.3390/ijms24119361

**Published:** 2023-05-27

**Authors:** Margo I. Jansen, Alessandro Castorina

**Affiliations:** Laboratory of Cellular and Molecular Neuroscience (LCMN), School of Life Sciences, Faculty of Science, University of Technology Sydney, P.O. Box 123, Broadway, NSW 2007, Australia; margo.jansen@student.uts.edu.au

**Keywords:** multiple sclerosis, white matter, microarray meta-analysis

## Abstract

Multiple sclerosis (MS) is an autoimmune disorder of the central nervous system (CNS) whose aetiology is only partly understood. Investigating the intricate transcriptional changes occurring in MS brains is critical to unravel novel pathogenic mechanisms and therapeutic targets. Unfortunately, this process is often hindered by the difficulty in retrieving an adequate number of samples. However, by merging data from publicly available datasets, it is possible to identify alterations in gene expression profiles and regulatory pathways that were previously overlooked. Here, we merged microarray gene expression profiles obtained from CNS white matter samples taken from MS donors to identify novel differentially expressed genes (DEGs) linked with MS. Data from three independent datasets (GSE38010, GSE32915, and GSE108000) were combined and used to detect novel DEGs using the Stouffer’s Z-score method. Corresponding regulatory pathways were analysed using the Gene Ontology (GO) and Kyoto Encyclopedia of Genes and Genomes (KEGG) pathway databases. Finally, top up- and down-regulated transcripts were validated by real-time quantitative PCR (qPCR) using an independent set of white matter tissue samples obtained from MS donors with different disease subtypes. There were a total of 1446 DEGs, of which 742 were up-regulated and 704 genes were down-regulated. DEGs were associated with several myelin-related pathways and protein metabolism pathways. Validation studies of selected top up- or down-regulated genes highlighted MS subtype-specific differences in the expression of some of the identified genes, underlining a more complex scenario of white matter pathology amongst people afflicted by this devastating disease.

## 1. Introduction

Multiple sclerosis (MS) is a demyelinating disease of the central nervous system (CNS), characterised by chronic inflammation and neurodegeneration [1,2]. In MS, oligodendrocytes, the myelin-producing cells of the CNS, are damaged, resulting in the progressive loss of myelin and consequent formation of multi-focal lesions within the white matter (WM). The exact pathogenesis of MS is unknown; however, it is believed to be the result of a complex interaction between environmental, genetic, and lifestyle factors [3].

The clinical course of MS is categorised into three main subtypes: relapse-remitting MS (RRMS), secondary-progressive MS (SPMS), and primary-progressive MS (PPMS). Each subtype follows its own clinical course, with RRMS usually presenting with recurrent relapses of symptoms followed by periods of relative stability and recovery [4]. It is common for patients diagnosed with RRMS to progress into SPMS (~80% of cases), where the episodes of remission no longer occur, and progressive worsening of clinical symptoms ensue [5]. The third main subtype of MS is PPMS, a rare form of MS that only represents around 10% of diagnosed patients [6]. This form of MS is characterised by the progressive worsening of symptoms from the onset of the disease.

Lesions in the WM of MS patients can either be defined as active, inactive, chronic active or inactive, and regenerating based on their histopathological characteristics [7,8]. The formation of these pathological demyelinating lesions of the WM is accompanied—and often exacerbated—by co-existing factors such as inflammation, neurodegeneration, and gliosis [7,8,9,10,11]. However, it is not uncommon to observe signs of remyelination in some lesion types [12]. Despite current efforts, the exact aetiology of the disease remains unknown, and many questions on the molecular mechanisms behind lesion development remain obscure. Interestingly, the WM surrounding lesions is often devoid of any obvious signs of injury and is nearly indistinguishable from the WM tissue of healthy subjects. However, molecular studies suggest that certain abnormalities, such as altered tight junctions and early signs of inflammation, can still be detected along the rims of lesions and in the peri-lesional WM [13,14,15]. Thus, investigating the transcriptional profile of WM samples gathered from different regions and with different pathophysiological characteristics could be useful for the identification of transcriptomic signatures of disease in spite of the heterogeneity of tissue and clinical presentation.

Researchers have used several methods to decipher pathological changes in the WM of people afflicted with MS. With the advancement of high-throughput techniques such as microarray and RNA-sequencing, scientists have attempted to explore the transcriptomic profile of the WM of MS patients [16,17,18]. Whilst these approaches have helped to gain some insights into gene expression changes across different lesion types, key factors such as small sample size, variations in storage conditions, and quality of samples RNA (among others) have hindered the depth of these analyses. Hence, in this study, we merged multiple publicly available datasets with the idea of overcoming some of these issues whilst also subserving the need to provide a more in-depth evaluation of available transcriptomic data [19,20,21].

To achieve this goal, we performed a meta-analysis using transcriptomic data obtained from three publicly available microarray datasets obtained from WM samples collected from MS donors and aged-matched non-MS controls. In addition, to validate our findings and assess whether gene expression changes were consistent across the different MS subtypes, we performed reverse transcription quantitative real-time PCR (RT-qPCR) of selected top novel up- and down-regulated transcripts using WM samples from aged-matched non-MS and MS donors (RRMS, SPMS, and PPMS) readily available in our laboratory.

## 2. Results

### 2.1. Data Collection and Meta-Analysis

After searching the GEO database for studies that met our inclusion-exclusion criteria (see Section 4.1), three datasets were selected for the meta-analysis study (please refer to the experimental workflow shown in Figure 1).

For each dataset, we grouped sample data into non-MS and MS WM tissues to allow comparisons between non-pathological and pathological tissue, irrespective of lesion type or WM appearance. This approach allowed us to maximise the number of samples included in the meta-analysis, hence providing a larger cohort of samples from which we could extrapolate more in-depth transcriptional changes. The GSE38010 dataset contained 5 MS samples and 2 control samples. GSE32915 contained 12 MS samples and 4 control samples, while GSE108000 contributed 30 MS samples and 10 control samples. Pre-processing of the three datasets resulted in a comparable range of expression levels (Appendix A).

Following pre-processing of datasets, we utilised Stouffer’s Z-score method to determine the differentially expressed genes (DEGs) in our meta-analysis. Evidence has shown that this method outperforms Fisher’s combined probability test, which has been a popular method for microarray meta-analyses [21,22]. Using the former probability test, we found that 1446 genes were differentially expressed when comparing control versus MS WM samples (please see Appendix A). A total of 742 genes were significantly up-regulated (log2 Fold Change (FC) ≥ 0.5 in the combined dataset, and 704 genes were significantly down-regulated (log2 (FC) ≤ −0.5. An overview of the top 15 up- and down-regulated DEGs are shown in Table 1 below.

Using the gene set variation analysis [23], we also aimed to replicate our findings by comparing DEGs from our merged microarray dataset with DEGs obtained from a publicly available RNA-sequencing dataset. Following the comparison with GSE138614, we confirmed that the general transcriptomic alterations observed in the merged dataset were consistent with those in the RNA-sequencing dataset (Appendix A).

### 2.2. Functional Classification and Pathway Analyses of DEGs

To gather functional insights into the biological and intracellular processes associated with the DEGs obtained from our meta-analysis, we utilised Gene Ontology (GO) and Kyoto Encyclopedia of Genes and Genomes (KEGG) databases. The top 15 annotated GO terms and KEGG pathways are depicted in Figure 2A,B.

Both analyses allow the extrapolation of the biological pathways and intracellular mechanisms that are altered based on the number and statistical power of the annotated DEGs from our combined analysis. The most significant GO term was *central nervous system myelination* (FDR = 9.87 × 10^−5^). As expected from a disease affecting the WM, most of the top GO terms were associated with oligodendrocytes- or myelin-related alterations. Interestingly, we also identified DEGs that were linked to protein metabolism/turnover (*proteasome-mediated ubiquitin-dependent protein catabolic process*, FDR = 2.98 × 10^−3^; *negative regulation of protein phosphorylation*, FDR = 3.078 × 10^−3^; and *regulation of proteasomal ubiquitin-dependent protein catabolic process*, FDR = 6.870 × 10^−3^).

In contrast, KEGG analysis of the DEGs from the merged dataset revealed no statistically significant associations with any specific intracellular pathway (Figure 2B). However, some relevant pathways emerged as being affected by the disease, such as those pertaining to insulin resistance (FDR = 0.0977), axon guidance (FDR = 0.0977), and cell adhesion molecules (FDR = 0.162).

### 2.3. Unique DEGs Detected from the Meta-Analysis

Following our analyses of merged datasets, we sought to determine if the unified dataset improved the depth of the analysis, enabling us to identify unique DEGs that could not be identified in the three datasets when analysed individually. Using the limma package, we performed DEG analyses comparing the Control versus MS tissue of GSE32915, GSE38010, and GSE108000 individually (Appendix A). The overlap between DEGs found in each analysis and our meta-analysis is illustrated in Figure 3.

As it can be appreciated in Figure 3, whereas each dataset produced several DEGs (GSE380010 = 663, GSE32915 = 405, and GSE108000 = 3830, respectively), only four DEGs were in common among the three individual datasets, and the number increased to nine DEGs when the combined dataset was also included. Interestingly, the combined meta-data identified 175 new DEGs (Appendix A). Table 2 shows the top 15 up- and down-regulated genes extrapolated from the combined dataset.

GO analyses of the unique 175 DEGs obtained from the combined dataset revealed WM alterations pertaining to G-protein-coupled receptor signalling pathways (Figure 4A). Using the KEGG database, terms associated with lipid/fat metabolism dominated amongst the top 15 associated terms, including *fatty acid degradation* (FDR = 3.177 × 10^−18^), *regulation of lipolysis in adipocytes* (FDR= 1.580 × 10^−10^), and *fatty acid metabolism* (FDR = 3.554 × 10^−8^). Other annotated terms that were significantly different were linked to pyruvate metabolism (FDR = 6.331 × 10^−13^) and cyclic guanosine monophosphate (cGMP)-protein kinase G (PKG) signalling (FDR = 6.658 × 10^−12^).

### 2.4. Validation of Selected DEGs Using Real-Time Quantitative PCR

Since meta-analyses are statistical tools that rely on *p*-value combination methods, we thought it was of interest to determine whether selected DEGs that were uniquely affected in the unified dataset could be validated by real-time quantitative PCR (RT-qPCR). For this purpose, we used an independent set of WM tissues obtained from MS donors that had been previously diagnosed with different clinical MS subtypes (i.e., relapse-remitting [RRMS], secondary-progressive [SPMS], and primary-progressive MS [PPMS]) and non-MS age-matched controls.

Upon evaluation of the top 15 up- and down-regulated DEGs detected in our meta-analysis (Table 2), we selected eight genes (four up-regulated and four down-regulated) for validation. Genes were selected based on their association with brain pathology, myelin cell function, and autoimmunity [24,25,26,27,28,29,30,31,32,33,34]. Selected genes were: Bardet-Biedl syndrome-7 *(BBS7),* Epoxide hydrolase 4 *(EPHX4),* L1 Cell Adhesion Molecule *(L1CAM),* nuclear factor of activated T cells 5 *(NFAT5),* Cytochrome B5 Reductase 2 *(CYB5R2),* protein tyrosine phosphatase 4A1 *(PTP4A1),* G Protein-Coupled Receptor-37 *(GPR37),* and Very Low-Density Lipoprotein Receptor *(VLDLR)*.

Surprisingly, *BBS7* gene expression was not different in the WM of MS versus non-MS samples (Figure 5A), although when results were clustered based on MS subtype, data showed a bimodal distribution of data for RRMS cases and a trend towards an increase in expression for SPMS cases (Figure 5A′). Analyses of *EPHX4* transcripts demonstrated increased gene expression in the MS group, although these were not statistically significant (*p* = 0.089 vs. control; Figure 5B). However, upon clustering data by clinical subtypes, SPMS-specific *EPXH*4 gene up-regulation was found to contribute remarkably to the overall up-regulation seen in MS samples (** *p* < 0.01 vs. control; Figure 5B′), with a minor increase in gene expression also identified in RRMS cases (*p* > 0.05 vs. control). Similarly, *L1CAM* expression was increased in MS samples versus non-MS controls, although not achieving statistical significance (*p* = 0.0771; Figure 5C). At a subtype-specific level, this trend towards an increase was driven solely by a robust up-regulation in SPMS cases (*** *p* < 0.001; Figure 5C′) and marginally contributed by PPMS, but not RRMS cases. Finally, an overall and significant increase in *NFAT5* gene expression levels was seen in all MS cases (* *p* < 0.05; Figure 5D), which was contributed by RRMS (*p* = 0.051; Figure 5D′) and SPMS (** *p* < 0.01; Figure 5D′), but not PPMS cases (*p* > 0.05; Figure 5D′).

Upon examining the expression of genes predicted to be down-regulated, we observed more robust effects (Figure 5). RT-qPCR validation of *CYB5R2*, a gene encoding for a protein involved in cholesterol biosynthesis, showed a strong reduction in expression levels among MS samples when compared with non-MS controls (*** *p* < 0.001; Figure 5E), an effect that was visible across all the tested MS subtypes (* *p* < 0.05 for RRMS, ** *p* < 0.01 for SPMS and PPMS, respectively; Figure 5E′). *PTP4A1* mRNAs also confirmed the down-regulation seen in our bioinformatics studies (* *p* < 0.05 vs. controls; Figure 5F). However, the subtype-specific down-regulation was statistically significant in RRMS cases (** *p* < 0.01; Figure 5F′) and only marginally reduced in progressive MS subtypes (*p* > 0.05). A strong and global *GPR37* down-regulation was observed in the MS WM (**** *p* < 0.0001; Figure 5G), which was consistently seen across all MS subtypes (**** *p* < 0.0001 for RRMS, *** *p* < 0.001 for SPMS and PPMS, respectively; Figure 5G′). Similarly, *VLDLR* gene expression was also remarkably reduced in the MS WM (*** *p* < 0.0001; Figure 5H), and the reduction was equally contributed when examined at a subtype-specific level (** *p* < 0.01 for both RRMS, SPMS, and PPMS; Figure 5H′).

## 3. Discussion

In the present study, we conducted a meta-analysis by merging three publicly available microarray datasets; the final goal was to exploit this combinatorial approach to unveil additional transcriptomic changes in the MS WM that may have been overlooked when performing individual analyses using single datasets. Indeed, whilst the meta-analysis identified a number of genes that overlapped with those from the individual datasets, it revealed several new DEGs that were not previously reported. Moreover, our validation studies on a panel of selected genes whose expression levels were uniquely altered in our combined dataset confirmed the predicted gene expression changes. For this purpose, we utilised an independent set of WM samples from donors with different clinical forms of MS and age-matched controls, thus enabling us to unveil additional transcriptional information pertaining to MS subtype-specific alterations amongst the newly identified DEGs (Figure 5).

GO and KEGG analyses of combined metadata demonstrated that, in addition to the expected annotation of terms linked to oligodendrocyte differentiation, myelination, and oligodendrocyte development, perturbations in biological pathways associated with lipid and protein metabolism were also detected. Impaired lipid metabolism within the WM has been associated with microglia senescence [35], hindered myelin maintenance [36], and reduced regeneration capacity [37] in both rodent and human studies. Therefore, results from our bioinformatics approach further confirmed such alterations of these biological processes in human ex vivo MS WM tissues. Similarly, a systematic review of proteomic studies reported several imbalances of WM proteostasis in MS brains [38], as seen in our study, although these reports showed some inconsistencies that were most likely due to the clinical heterogeneity of the disease. KEGG analyses also revealed alterations in pathways associated with insulin resistance. While the strength of this association was not statistically significant, it raises important questions on the possible relationship between systemic glucose sensitivity and MS. In this regard, studies have shown that insulin resistance has a high predictive value for disease severity in people with MS, especially among those with a diagnosis of SPMS [39,40]. Additionally, insulin resistance was found to exert detrimental effects on oligodendrocyte and myelin health in other neurodegenerative conditions such as Alzheimer’s disease [41], multiple system atrophy [42], and in developmental pathologies caused by ethanol consumption, such as foetal alcohol spectrum disorder [43]. These reports provide more than a simple anecdotal evidence of the link existing between glucose metabolism and WM health and warrant further investigations to help better understand the crosstalk between insulin signalling and the functionality of myelin-producing cells.

Through comparative analyses of DEGs from the merged dataset and the individual source datasets, we were able to define a subset of novel DEGs. Enrichment (GO) pathway analyses of this subset of altered genes revealed robust and significant links with G-protein-coupled receptor (GPCR) signalling pathways. GPCR signalling plays a critical role in regulating both the maturation of myelin cell and immune cell activities. For instance, intrinsic signalling of the G-protein-coupled receptor 17 (GPR17) in oligodendrocyte progenitors was found to inhibit different stages of cell maturation [44]. Other well-established and relevant GPCRs linked to autoimmunity and MS are the sphingosine-1-phosphate receptors [45,46], which are specifically targeted by the best available drugs approved for the treatment of relapsing and secondary progressive MS forms, such as fingolimod [47] and siponimod [48].

From KEGG analyses of novel DEGs uniquely found in the merged dataset, we also identified robust alterations of pathways associated with fatty acid degradation. The disruption of lipid metabolism due to impaired degradation of fatty acids is detrimental to myelin formation and stability. Myelin is a modified cell membrane that forms a multilayer sheath around the axon. Whilst myelinated sheaths retain several features of conventional cell membranes, myelin constituent lipid components differ, and so do fatty acids [49]. Indeed, there is evidence from mutations associated with demyelination to suggest that myelin health requires intact lipid pathways with a steady balance between lipid synthesis and degradation [50,51]. As such, data from our bioinformatics analyses brings under the spotlight the critical role of lipid pathways in the regulation of myelin synthesis and repair, pinpointing how these underlying WM defects play a key role in the onset and progression of MS pathology.

To strengthen the bioinformatics analyses, selected DEGs that appeared exclusively in our merged dataset were validated by RT-qPCR using WM tissue samples obtained from MS donors with different disease subtypes (RRMS, SPMS, or PPMS).

Two genes associated with cholesterol/lipid metabolism appeared amongst the top 15 of the 175 newly detected DEGs, namely *EPHX4* and *VLDLR*, with *EPHX4* being up-regulated in SPMS cases and *VLDLR* being down-regulated in all subtypes of MS [26,34]. Impaired lipid metabolism has been linked to disease progression in MS and was found to have strong predictive value in determining the degree of neurodegeneration and lesions’ formation [52,53,54,55,56]. Cholesterol is an essential lipid for the synthesis of myelin, and aberrant cholesterol metabolism is linked to both impaired remyelination and myelin-debris phagocytosis [37,57,58,59,60]. Interestingly, another of our selected genes for validation that was down-regulated (*GPR37*) is linked to oligodendrocyte functioning. GPR37 is a GPCR whose expression is normally increased in pre-myelinating and myelinating oligodendrocytes and acts as a negative regulator of oligodendrocyte differentiation [29]. In fact, *GPR37*-null mice exhibit signs of hypermyelination. The strongly reduced expression of this gene we observed in our meta-analysis and validation experiments may suggest the inherent need to increase the maturation of surviving oligodendrocytes in the WM distal to the lesion site of MS patients or alternatively, promote the phenotypic transition from oligodendrocyte precursors to mature oligodendrocytes cells, perhaps in the attempt to replenish the depleting pool of myelinating cells.

However, the pathological processes that underpin MS disease are not limited to demyelination and/or impaired remyelination. Altered expression of other molecular targets, such as cell-adhesion molecules, has also been implicated in disease pathophysiology, as these have a strong impact on immune-cell functioning, maintenance of blood-brain barrier integrity, and myelin density [13,14,61,62,63,64,65]. *L1CAM* is a gene that encodes for a cell-adhesion molecule that we found to be significantly up-regulated in the WM of SPMS patients only. Although it is not essential for ongoing myelination, L1CAM plays an important role in the initiation of myelination and supports the interactions between axons and oligodendrocytes, a process that appears to be activated upon the proteolysis of the L1CAM protein [31,66,67]. Since L1CAM proteolytic products (L1 molecules) have been detected in brain lysates, the increased expression of L1CAM in SPMS cases suggests the increased need for cleaved L1CAM proteins to restore homeostasis in patients with this progressive form of MS [68].

Altogether, the results reported here highlight a complex scenario where different pathological processes and intracellular pathways coexist and prevail over homeostatic mechanisms, culminating in a reduced WM regeneration potential, especially in SPMS.

Notwithstanding that this study provided novel transcriptomic data, it is noteworthy mentioning that the meta-analysis included only three publicly available microarray datasets. This limited our ability to balance the number of controls versus MS samples. Although, in principle, this could introduce a small risk of sampling bias, our validation experiments confirmed that the newly detected genes from our meta-analysis were indeed significantly deregulated in the MS WM, thus providing new molecular targets to investigate in the context of MS.

Taken together, this meta-analysis revealed additional information about gene expression changes in the WM of MS donors, irrespective of lesion type or tissue appearance. After merging the datasets, we were able to provide additional DEGs that were not detected previously in the individual datasets. Additionally, validation studies proved that the expression of selected genes matched that predicted in our meta-analysis, further highlighting the strength of bioinformatics tools for the identification of novel genes and regulatory pathways to gain a better understanding of MS and identify new molecular targets for therapeutic development.

## 4. Materials and Methods

### 4.1. Datasets Selection and Pre-Processing of Data

The public database Gene Expression Omnibus (GEO) was used to search for available micro-array datasets obtained from MS WM (and non-MS controls). All the available microarray studies investigating human MS brain WM tissue were considered. Each dataset had to contain control brain samples from age-matched individuals to be included in the meta-analysis. No distinctions were made between MS subtypes or the typology of the lesion in this meta-analysis. Moreover, no distinction between microarray platforms was made considering the limited number of studies available. Based on these criteria, three datasets from GEO were included in this meta-analysis, namely [GEO:GSE108000], [GEO:GSE32915], and [GEO:GSE38010].

The following steps were all performed in RStudio (Version 1.4.1717; R version: 4.1.1). Using the R package GEOquery, raw data and sample metadata from the aforementioned datasets were obtained [69]. The Affymetrix dataset, GSE38010, was normalised using “Robust-Multi array Average” normalisation with the use of the affy package [70]. Raw data for the remaining two datasets, GSE32915 and GSE108000, both performed on the Agilent platform, were read and normalised using the limma package [71]. In the case of GEO:GSE108000, only the channel containing experimental data was included to keep normalisation steps similar among datasets. Both background corrections and quantile between-array normalisations were performed on these two datasets. The datasets were then annotated using the appropriate annotation packages obtained from Bioconductor.

### 4.2. Meta- and Differential Gene Expression Analysis

Meta-analysis and subsequent analyses of DEGs were performed using the recently developed DExMA package [72]. Briefly, the meta-analysis was performed on the normalised datasets using Stouffer’s Z-score method, comparing MS WM tissue with non-MS control samples [21]. Genes had to be present in all three datasets for the gene to be included in the meta-analysis. The identified (DEGs) were considered statistically significant if the log2 fold change (FC) was > 0.5 or < −0.5 and if the false discovery rate was < 0.05. Individual DEG analyses of each dataset were performed using the limma package according to the manual’s instructions.

### 4.3. Gene Ontology and Pathway Analysis

Gene Ontology (GO) classification and Kyoto Encyclopedia of Genes and Genomes (KEGG) pathway analyses were performed using the DEGs obtained from comparing non-MS brain WM tissue with MS WM tissues. To achieve these analyses, both the GO and KEGG analyses were implemented in RStudio using the “ClusterProfiler” package [73].

### 4.4. Gene set Variation Analysis

In RStudio (Version 1.4.1717; R version: 4.1.1), a gene set variation analysis (GSVA) was performed using the package ‘GSVA’ (1.42.0, Bioconductor) using the publicly available RNA-sequencing dataset GSE138614 [16,23]. Briefly, the raw count matrix was obtained from the NCBI GEO database and transformed using the variance stabilizing transformation (VST) algorithm [74]. Either up- or down-regulated DEGs detected in our meta-analysis were used for the analysis. Results were plotted using GraphPad Prism (version 9.3.1 for Windows, GraphPad Software, San Diego, CA, USA, www.graphpad.com; last accessed on 31 March 2023).

### 4.5. Human Post-Mortem Brain Tissue

CNS WM samples from MS and non-MS donor tissues were obtained from the MS Research Australia Brain Bank (Tissue Transfer Deed—CT31920, approved on 21 June 2021) and the Victorian Brain Bank (Material Transfer Agreement—VBB.19.07, approved on 16 January 2020). In total, 16 samples from RRMS, SPMS, and PPMS donors were used for validation studies. As a control, we used additional tissue shavings obtained from non-MS age-matched donors. The *post-mortem* intervals varied between 15–32 h. Neuropathological assessments of MS tissues were conducted by Dr. Antony Harding and Dr. Andrew Affleck (MS Brain Bank, NSW, Australia), whereas those for non-MS samples were performed by Prof. Catriona McLean (Victoria Brain Bank, VIC, Australia). Demographic details of tissue donors are shown in Table 3.

### 4.6. RNA Isolation and Real-Time Quantitative PCR

Micro-dissections of snap-frozen WM shavings from both MS and non-MS donors were performed using a stereoscopic microscope (10× magnification) under RNase-free conditions. Briefly, dissected tissue shavings (~30–40 mg wet-weight) were immediately submerged in RNAlater™ Stabilization Solution (Thermo Fisher Scientific, Scoresby, VIC, Australia) and then snap-frozen in liquid nitrogen until further processing. Total RNA was extracted using TRIreagent (Sigma-Aldrich, Castle Hill, NSW, Australia). Briefly, samples were homogenised using γ-irradiated autoclavable pestles (Sigma-Aldrich, Castle Hill, NSW, Australia) and then centrifuged at 12,000 × *g* at 4 °C, 15 min in the presence of 200 µL chloroform (Sigma-Aldrich, Castle Hill, NSW, Australia). The RNA fraction was collected and spun down with 500 µL ice-cold 2-propanol (Sigma-Aldrich, Castle Hill, NSW, Australia) as above. Pellets containing RNA were then washed twice with 75% ethanol and air-dried. Thereafter, RNA was subjected to DNAse I treatment (Thermo Fisher Scientific, Scoresby, VIC, Australia), followed by a clean-up step using the RNeasy Micro Kit (Qiagen, Clayton, VIC, Australia). RNA concentrations were determined using a NanoDrop™ 2000 spectrophotometer (Thermo Fisher Scientific, Scoresby, VIC, Australia).

cDNA was generated using the Tetro cDNA synthesis kit (Bioline, Sydney, NSW, Australia) according to the manufacturer’s instructions. To analyse mRNA expression, we performed real-time quantitative PCRs for a selection of eight genes (the primers list is shown in Table 4). Each reaction contained 5 μL iTaq Universal SYBR Green Supermix (Bio-Rad, South Granville, NSW, Australia), 3μL of cDNA (100 ng), and 0.8 µL forward and reverse primers (final concentration = 500 nM). Ribosomal protein S18 was used as a reference gene. For the analysis, mean fold changes were calculated using the ΔΔCt method, as described previously [75].

## Figures and Tables

**Figure 1 ijms-24-09361-f001:**
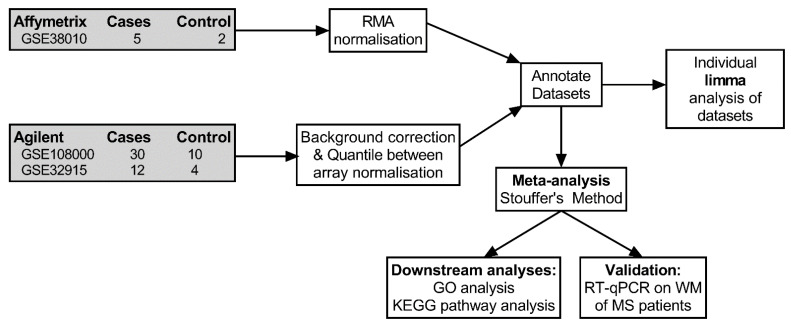
Experimental workflow. Workflow showing datasets collection, pre-processing, merging, and analyses to determine differentially expressed genes (DEGs) used for biological (GO), intracellular pathway analyses (KEGG), and validation.

**Figure 2 ijms-24-09361-f002:**
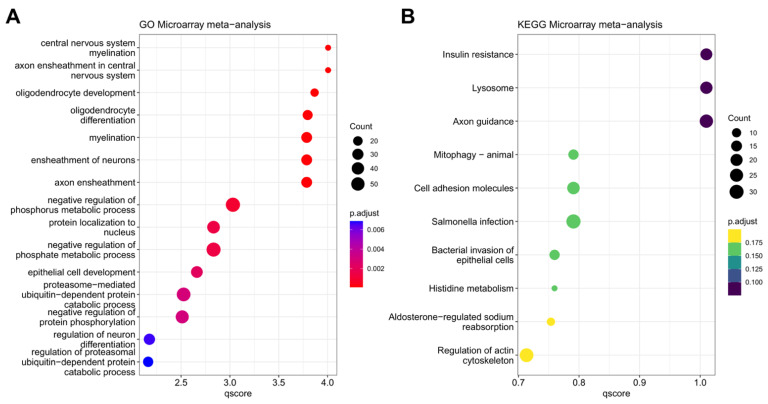
GO and KEGG pathway enrichment analyses of DEGs from meta-analysis. (**A**) Top 15 Gene Ontology terms and the (**B**) top 15 KEGG enriched pathways associated with the differentially expressed genes are shown. On the x-axis, the minimum −Log10 of the calculated adjusted *p*-value is given (qscore). On the y-axis, the associated GO (**A**) or KEGG pathway (**B**) terms are listed.

**Figure 3 ijms-24-09361-f003:**
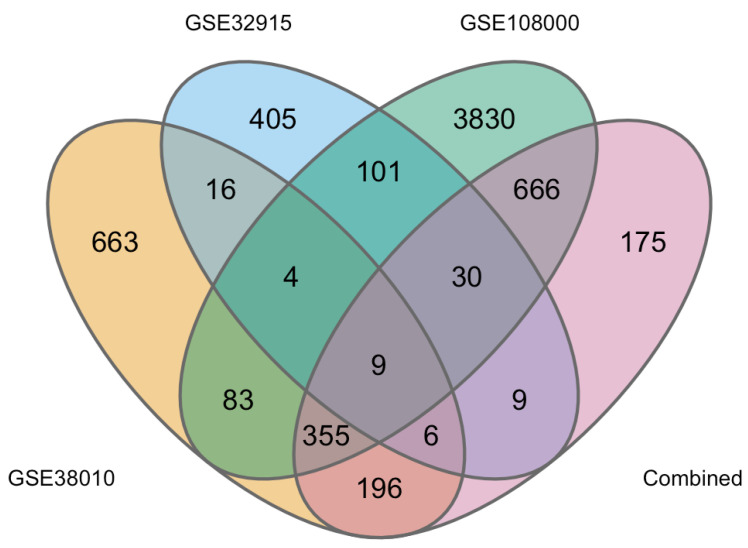
Venn diagram showing the overlap of DEGs found in each individual limma analysis and in the meta-analysis of the merged datasets. Overlapping DEGs identified the following individual limma analyses of GSE38010 (yellow), GSE32915 (blue), and GSE108000 (green), and in our combined dataset (pink) are shown.

**Figure 4 ijms-24-09361-f004:**
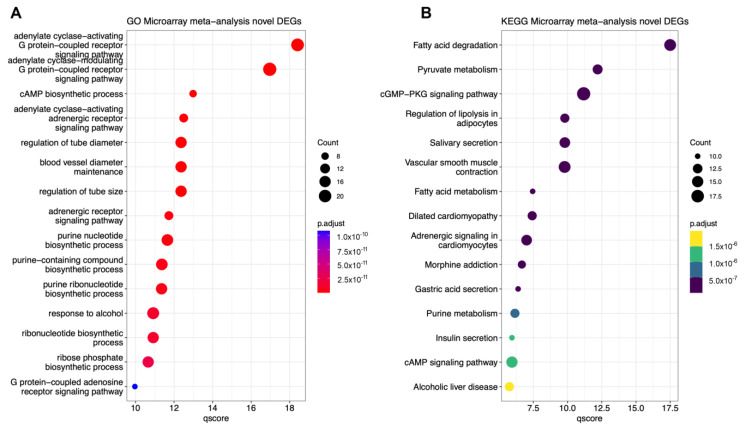
GO and KEGG analyses of novel DEGs obtained from the meta-analysis. (**A**) Top 15 Gene Ontology terms and the (**B**) top 15 KEGG enriched pathways associated with the differentially expressed genes. On the x-axis, the minimum −Log10 of the calculated adjusted *p*-value is given (qscore). On the y-axis, the associated GO (**A**) or KEGG pathway (**B**) terms are given.

**Figure 5 ijms-24-09361-f005:**
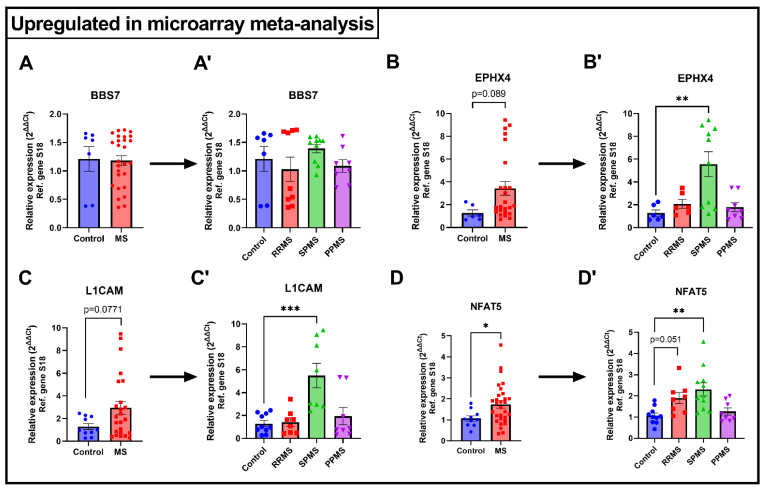
Validation of selected newly identified DEGs obtained from the meta-analysis. Real-time quantitative PCRs in WM samples from different MS clinical entities was used to validate gene expression changes that were predicted to be up-regulated (**A**,**A’**) *BBS7*, (**B**,**B’**) *EPHX4*, (**C**,**C’**) *L1CAM*, (**D**,**D’**) *NFAT5* or down-regulated (**E**,**E’**) *CYB5R2*, (**F**,**F’**) *PTP4A1*, (**G**,**G’**) *GPR37,* and (**H**,**H’**) *VLDLR* in Control vs MS. Bar graphs shown in A’ – H’ depict gene expression changes according to the different MS subtypes (RRMS, SPMS or PPMS). Fold changes were calculated with the ΔΔCt method using the expression of ribosomal protein S18 as a reference gene. *N* = 6–10 per group, with MS containing the combined results of each subtype. Mean ± SEM is plotted. * *p* < 0.05, ** *p* < 0.01, *** *p* < 0.001, **** *p* < 0.0001, as determined by unpaired *t*-test (control vs. MS) or one-way ANOVA followed by Dunnett’s post-hoc test (control vs. MS subtypes).

**Table 1 ijms-24-09361-t001:** Top 15 significantly up- and down-regulated genes in white matter tissue of non-MS versus MS donors.

Top 15 Up-Regulated Genes				
Symbol	Stat	Pval	FDR	AveFC
*DOK6*	2.159879	1.54 × 10^−2^	4.91 × 10^−2^	1.669468
*GPNMB*	4.065861	2.39 × 10^−5^	3.15 × 10^−4^	1.58023
*HOMER1*	2.621306	4.38 × 10^−3^	1.85 × 10^−2^	1.524352
*SNX10*	3.492226	2.40 × 10^−4^	1.87 × 10^−3^	1.509591
*FSTL5*	2.942985	1.63 × 10^−3^	8.52 × 10^−3^	1.453001
*ATRX*	5.076691	1.92 × 10^−7^	8.18 × 10^−6^	1.418288
*IGHM*	2.982941	1.43 × 10^−3^	7.68 × 10^−3^	1.402958
*STK38L*	5.544933	1.47 × 10^−8^	1.31 × 10^−6^	1.383691
*RB1CC1*	4.329887	7.46 × 10^−6^	1.28 × 10^−4^	1.364945
*PLPPR4*	2.503236	6.15 × 10^−3^	2.41 × 10^−2^	1.350802
*TRHDE*	2.961552	1.53 × 10^−3^	8.14 × 10^−3^	1.333155
*ZNF184*	3.273974	5.30 × 10^−4^	3.51 × 10^−3^	1.320123
*VPS13A*	2.325504	1.00 × 10^−2^	3.52 × 10^−2^	1.303272
*PLCXD3*	3.653899	1.29 × 10^−4^	1.16 × 10^−3^	1.294933
*ESF1*	6.202506	2.78 × 10^−10^	7.22 × 10^−8^	1.293416
**Top 15 Down-regulated genes**				
**Symbol**	**Stat**	**Pval**	**FDR**	**AveFC**
*CARNS1*	4.544461	2.75 × 10^−6^	6.09 × 10^−5^	−1.96092
*LDB3*	5.285667	6.26 × 10^−8^	3.69 × 10^−6^	−1.79909
*TMEM63A*	5.921268	1.60 × 10^−9^	2.55 × 10^−7^	−1.77689
*MAG*	4.884586	5.18 × 10^−7^	1.70 × 10^−5^	−1.74786
*GPIHBP1*	4.063876	2.41 × 10^−5^	3.17 × 10^−4^	−1.67791
*CMTM5*	4.830894	6.80 × 10^−7^	2.09 × 10^−5^	−1.64751
*ZFYVE16*	2.623818	4.35 × 10^−3^	1.84 × 10^−2^	−1.62818
*MOG*	3.347922	4.07 × 10^−4^	2.85 × 10^−3^	−1.60904
*OLIG2*	6.949123	1.84 × 10^−12^	1.65 × 10^−9^	−1.59546
*CNDP1*	2.652296	4.00 × 10^−3^	1.72 × 10^−2^	−1.59361
*CDK18*	7.56455	1.95 × 10^−14^	1.14 × 10^−10^	−1.58967
*SLC5A11*	2.697006	3.50 × 10^−3^	1.56 × 10^−2^	−1.58249
*KLK6*	3.218194	6.45 × 10^−4^	4.11 × 10^−3^	−1.57037
*ELOVL1*	3.35047	4.03 × 10^−4^	2.83 × 10^−3^	−1.56773
*CERCAM*	5.915865	1.65 × 10^−9^	2.62 × 10^−7^	−1.51354

Stat = statistical value calculated using Stouffer’s method, FDR = false discovery rate; Average FC = average log2 fold change. Genes were considered differentially expressed if the FDR < 0.1 and if the average log2 (FC) > 0.5 or < −0.5.

**Table 2 ijms-24-09361-t002:** Top 15 newly identified DEGs obtained from the meta-analysis.

Top 15 Up-Regulated Genes				
Symbol	Stat	Pval	FDR	AveFC
BBS7	3.507815	2.26 × 10^−4^	0.001787	1.089201
RGS17	2.898288	1.88 × 10^−3^	0.009556	1.042207
TNRC6B	2.655782	3.96 × 10^−3^	0.017099	1.034418
STYK1	2.775744	2.75 × 10^−3^	0.012905	1.031931
EPHX4	2.36644	8.98 × 10^−3^	0.032357	0.999705
L1CAM	2.840294	2.25 × 10^−3^	0.011056	0.911055
MFSD4A	2.591709	4.78 × 10^−3^	0.019783	0.909036
UFL1	3.487636	2.44 × 10^−4^	0.001901	0.90871
NFAT5	3.960667	3.74 × 10^−5^	0.000442	0.880825
IER5L	2.431444	7.52 × 10^−3^	0.028272	0.876896
THAP5	3.578819	1.73 × 10^−4^	0.00146	0.874155
LRATD1	2.224882	1.30 × 10^−2^	0.043078	0.858158
IFT57	2.995845	1.37 × 10^−3^	0.007434	0.852161
DMXL2	2.36112	9.11 × 10^−3^	0.032689	0.834282
ATP2B1	2.917813	1.76 × 10^−3^	0.0091	0.805719
**Top 15 Down-regulated genes**				
**Symbol**	**Stat**	**Pval**	**FDR**	**AveFC**
CYB5R2	2.254104	1.21 × 10^−2^	4.06 × 10^−2^	−1.20986
VGLL1	2.62897	4.28 × 10^−3^	1.82 × 10^−2^	−1.16498
SNORC	2.700465	3.46 × 10^−3^	1.54 × 10^−2^	−1.06094
SS18	3.225204	6.29 × 10^−4^	4.03 × 10^−3^	−1.02363
PTP4A1	2.885216	1.96 × 10^−3^	9.88 × 10^−3^	−1.01777
GPR37	2.553827	5.33 × 10^−3^	2.15 × 10^−2^	−1.0145
NAV2	4.894387	4.93 × 10^−7^	1.63 × 10^−5^	−0.93913
VLDLR	2.282063	1.12 × 10^−2^	3.84 × 10^−2^	−0.92436
YIF1A	4.376789	6.02 × 10^−6^	1.09 × 10^−4^	−0.92082
SLC35E3	2.611671	4.51 × 10^−3^	1.89 × 10^−2^	−0.8769
HSPA2	2.945238	1.61 × 10^−3^	8.48 × 10^−3^	−0.85283
DNAJC14	2.534753	5.63 × 10^−3^	2.25 × 10^−2^	−0.83403
TRIM27	3.574216	1.76 × 10^−4^	1.48 × 10^−3^	−0.82891
TYRO3	3.772104	8.09 × 10^−5^	8.02 × 10^−4^	−0.7971
GDE1	2.533789	5.64 × 10^−3^	2.25 × 10^−2^	−0.78212

Stat = statistical value calculated using Stouffer’s method; Pval = *p*-value; FDR = false discovery rate; Average FC = average log2 Fold-Change. Genes were considered differentially expressed if the FDR < 0.1 and if the average log2 (FC) > 0.5 or < −0.5.

**Table 3 ijms-24-09361-t003:** Demographic information of non-MS and MS donors. PMI = Post-mortem interval.

Group	Age (Years)	Place of Birth	Sex	PMI (Hours)	MS Duration (Years)	Lesion Type
Control	79	Australia	Female	59	N/A	N/A
Control	82	England	Female	25	N/A	N/A
Control	83	Australia	Male	27	N/A	N/A
Control	73	Australia	Male	22	N/A	N/A
Control	73	Australia	Female	26.5	N/A	N/A
RRMS	70	Australia	Male	21	43	Chronic active
RRMS	40	Australia	Male	5	8	Chronic active
RRMS	72	Australia	Female	31	20	Chronic active
RRMS	79	New-Zealand	Female	24	29.5	Chronic active
RRMS	82	Australia	Female	19	33.1	Chronic active—minimal regeneration
SPMS	57	Australia	Female	26.8	17.9	Chronic active—minimal regeneration
SPMS	68	Australia	Female	15	33.5	Chronic active—minimal regeneration
SPMS	69	New-Zealand	Female	8.5	38	Chronic active—minimal regeneration
SPMS	84	Australia	Female	15	42	Chronic active—minimal regeneration
SPMS	47	Australia	Female	20.8	25.8	Chronic active—minimal regeneration
SPMS	55	Australia	Male	7	40.1	Chronic active—minimal regeneration
PPMS	36	Australia	Female	24	13	Chronic active
PPMS	83	Australia	Female	16	16	Chronic active
PPMS	73	Australia	Male	25	15.6	Chronic active—moderate regeneration
PPMS	73	England	Male	24	41	Chronic active—minimal regeneration

**Table 4 ijms-24-09361-t004:** Primer sets used in RT-qPCR analyses.

Accession Number	Gene	Primer Sequence (5′-3′)	Length (bp)
NM_176824.3	*BBS7*	Fwd: CACATCTTGAAAGACTCTATGGC Rev: GCTGCATCGAAGAATGAAATCAA	151
NM_173567.5	*EPHX4*	Fwd: AGATGGCTGAAGTCACAAAGAT Rev: TCACTATGTCAGGTTGGTCTTG	102
NM_001278116.2	*L1CAM*	Fwd: AATTTGAGGACAAGGAAATGGC Rev: CTAAAGGTGTAGTGGACATAGGG	102
NM_138714.4	*NFAT5*	Fwd: GAGGACTTGCTGGATAACAGTC Rev: ATCATTGTAGGAACTGGTGCTC	135
NM_001302826.2	*CYB5R2*	Fwd: TGCCCTTGATTGAGAAAGAGAAA Rev: ATAGTTACCTACAGGAAGCCCTA	101
NM_003463.5	*PTP4A1*	Fwd: CTGTATTTGGAGAAGTATCGTCCT Rev: AGTTGTTTCTATGACCGTTGGA	70
NM_005302.5	*GPR37*	Fwd: GCCAAACTTGCTGTTATATGGG Rev: ATGGTGTCTGGTAAATCAGGAG	152
NM_022551.2	*S18*	Fwd: GAGGATGAGGTGGAACGTGT Rev: GGACCTGGCTGTATTTTCCA	115

## Data Availability

The micro-array datasets utilised in this study are publicly available and can be found here: GSE108000, GSE32915, and GSE38010. The RNA-sequencing dataset is available here: GSE138614. Any other data presented in this study can be made available upon reasonable request to authors.

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
