# Peer review of "Identification of Key Genes and Regulatory Pathways in Multiple Sclerosis Brain Samples: A Meta-Analysis of Micro-Array Datasets"

_ijms, 2023, doi:10.3390/ijms24119361_

Round 1

Reviewer 1 Report

In this study, Jansen and Castorina aimed to unrevail additional transcriptomic changes related to white matter pathology in MS. The topic of the research is very interesting given that the pathology of the disease could be associated not only with the demyelinated lesions but also with the NAWM environment. In the present study, authors combined analysis from publicly datashets from microarray and carry out the validation of candidate genes by RT-PCR. However a number of questions remain that should be addressed.

- Authors explain that they grouped sample data into non-MS and MS WM tissues to maximise the number of samples, comparing non-pathological and MS- tissue. In my opinion, the best option could be the comparision between WM from different MS clinical courses, i.e; RRMS, SPMS and PPMS. Given that the validation with RT-PCR was carried out by using the 3 different clinical courses, it could be interesting to analyse the publicly datasets in the same way to show additional transcriptomic changes related to white matter pathology in the different clinical courses of MS. I guess that it would be possible depending on the information provided by the publicly dataset.

-  In line 58 authors declare that “the WM surrounding these lesion sites is often devoid of any obvious histopathological signs of injury and is nearly indistinguishable from the white matter tissue of healthy subjects” . I do not agree with this sentence since WM surrounding the MS lesions is commonly named as normal appearing WM (NAWM) given that it seems to be control WM but in fact you can detect pathological characteristics such microglia activation. In sum, NAWM is far from normal in MS. Please, rephrase or justify with references.

- Demographic table with MS tissue should be improved, including more data such as the disease duration and, if possible, the type of MS lesions.

- It would be very useful a better explanation of the micro-dissection protocol. I mean, do you use slides or the snap-frozen blocks? Would be possible to characterize the MS lesions to improve the classification of each sample?

- The discussion about the results from GPR37 expression should be reconsidered. Given that GPR37 is considered as a negative regulator of oligodendrocyte differentiation, I do not understand the strongly down-regulation of this gene in MS samples. You might discuss the lack of information of gene expression related to different MS lesions since GPR37 could be up-regulated in those inactive MS lesions in which remyelination do not occur.

Minor comments:

- Line 33: use MS in brackets after the definition.

- Rephrase “In clinics” in line 40. Maybe, the sentence “The clinical courses of MS are categorized into three main subtypes” is more suitable.

- Relapse is commonly used to define the worsening of the symptoms instead of flare-ups (line 42).

- Please, use WM instead of the whole name white matter in line 52.

Some English words related to the MS field should be reconsidered.

Author Response

Response to comments from Reviewer #1
We would like to thank Reviewer #1 for his/her comments on our manuscript. We are happy to hear
the reviewer appreciates the benefit of these results for the field of Multiple sclerosis (MS) research
and how combining multiple datasets can provide novel insights into MS pathology. Please find our
responses to the comments below:
In this study, Jansen and Castorina aimed to unrevail additional transcriptomic changes related to
white matter pathology in MS. The topic of the research is very interesting given that the pathology of
the disease could be associated not only with the demyelinated lesions but also with the NAWM
environment. In the present study, authors combined analysis from publicly datashets from microarray
and carry out the validation of candidate genes by RT-PCR. However a number of questions remain
that should be addressed.
- Authors explain that they grouped sample data into non-MS and MS WM tissues to maximise the
number of samples, comparing non-pathological and MS- tissue. In my opinion, the best option could
be the comparision between WM from different MS clinical courses, i.e; RRMS, SPMS and PPMS.
Given that the validation with RT-PCR was carried out by using the 3 different clinical courses, it could
be interesting to analyse the publicly datasets in the same way to show additional transcriptomic
changes related to white matter pathology in the different clinical courses of MS. I guess that it would
be possible depending on the information provided by the publicly dataset.
RESPONSE: It was very pleasing to receive this comment, as this was our initial goal. Unfortunately,
upon analysis of clinical/demographic data from the few deposited datasets, only sparse clinical data
was offered and in some cases this information was completely omitted. Specifically, there were no
information regarding clinical diagnosis. This made it impossible for us to stratify data according to
disease subtype. However, we are fortunate to have post-mortem white matter samples obtained from
the Australian MS Brain Bank and were able to validate our data using these samples, allowing us to
provide additional evidence on the different expression levels of the identified genes when these were
clustered by MS subtype.
- In line 58 authors declare that “the WM surrounding these lesion sites is often devoid of any obvious
histopathological signs of injury and is nearly indistinguishable from the white matter tissue of healthy
subjects” . I do not agree with this sentence since WM surrounding the MS lesions is commonly named
as normal appearing WM (NAWM) given that it seems to be control WM but in fact you can detect
pathological characteristics such microglia activation. In sum, NAWM is far from normal in MS. Please,
rephrase or justify with references.
RESPONSE: We would like to thank the reviewer for noticing the inaccurate statement. We have now
rephrased the sentence in line 58 to better reflect the nature of normal-appearing white matter in MS.
Accordingly, the sentence was rephrased as follows: “…
- Demographic table with MS tissue should be improved, including more data such as the disease
duration and, if possible, the type of MS lesions.
RESPONSE: As recommended by this reviewer #1, we requested the Victorian and the MS Brain Bank
to provide additional information on the WM specimens used for validation. In the revised version of the
manuscript we included a more detailed table included such additional information. Thanks for raising
this issue.
- It would be very useful a better explanation of the micro-dissection protocol. I mean, do you use slides
or the snap-frozen blocks? Would be possible to characterize the MS lesions to improve the
classification of each sample?
RESPONSE: Thank to the reviewer for raising this point. We have added the word “snap-frozen” in line
450 to clarify that the tissues used for RNA extraction and real-time qPCR were fresh frozen (and not
fixed tissues). Based on our information from the MS Brain Bank, lesions contained within each tissue
block obtained from the different MS donors were all chronically active. However, it is important to note
that, as mentioned in the related Methods section, experiments were performed in the normal-appearing
white matter dissected from these tissue shavings (i.e. the most distal portion of WM from the lesion
site showing normal appearance macroscopically.
- The discussion about the results from GPR37 expression should be reconsidered. Given that GPR37
is considered as a negative regulator of oligodendrocyte differentiation, I do not understand the strongly
down-regulation of this gene in MS samples. You might discuss the lack of information of gene
expression related to different MS lesions since GPR37 could be up-regulated in those inactive MS
lesions in which remyelination do not occur.
RESPONSE: As mentioned in the discussion, given the negative regulatory activity of GPR37 in
oligodendrocyte differentiation and the previous observation of hypermyelination in GPR37-null mice,
we believe that (at least in normal-appearing WM) there might be a down-regulation to instigate
remyelination. To better clarify this point, the related paragraph was partly edited to avoid any confusion.
“…The strongly reduced expression of this gene we observed in our meta-analysis and validation
experiments may suggest the inherent need to increase maturation of surviving oligodendrocytes in the
WM distal to the lesion site of MS patients or alternatively, promote the phenotypic transition from
oligodendrocyte precursors to mature oligodendrocytes cells, perhaps in the attempt to replenish the
depleting pool of myelinating cells.
Minor comments:
- Line 33: use MS in brackets after the definition.
RESPONSE: updated.
- Rephrase “In clinics” in line 40. Maybe, the sentence “The clinical courses of MS are categorized
into three main subtypes” is more suitable.
RESPONSE: updated.
- Relapse is commonly used to define the worsening of the symptoms instead of flare-ups (line 42).
RESPONSE: updated.
- Please, use WM instead of the whole name white matter in line 52. –
RESPONSE: updated.
Demographic table with MS tissue should be improved, including more data such as the disease
duration and, if possible, the type of MS lesions.
RESPONSE: A new Table 3 containing additional information, including disease duration and type of
lesions was added in the revised version of the manuscript.

Reviewer 2 Report

The authors analyzed the combined data from three independent datasets (GSE38010, GSE32915 and GSE108000) by the Stouffer’s Z-score method, and found 1446 DEGs; 742 up-regulated and 704 down-regulated. The validation of the newly identified top up- and down-regulated DEGs by real-time quantitative PCR revealed the MS subtype-specific differences in the expression of these DEGs. Although the study results are interesting, I have the following comments.

 1. The qPCR results showed that the expression of the target genes varied according to the disease stage or subtype. This suggests that DEGs should be separately analyzed according to the disease stage or subtype. For instance, SPMS-specific DEGs could be quite different from those of RRMS.

 2. Why did the KEGG pathway analysis using DES in the combined GSE datasets show no statistically significant associations with any specific intracellular pathway?

 3. The average FCs of both the newly identified up- and down-regulated DEGs by the combined GSE datasets were much smaller compared with the combined GSE datasets. Why did the authors perform the GO and KEGG pathway analyses using only the newly identified DEGs? What is the rationale for this? These DEGs occupied the only small part of the total DEGs as determined by the combined GSE datasets. 

 4. As shown in Figure 3, the number of overlapped DEGs among each GSE dataset and the combined dataset was very small. It suggests a large variation of DEGs among the studies reflecting the disease heterogeneity, which makes the interpretation of the significance of the discovered DEGs very difficult.

 5. The qPCR data suggest that the newly discovered up-regulated DEGs are mostly related to SPMS. The authors should describe the relevant pathological changes in SPMS corresponding to the up-regulation of these DEGs. 

 6. What are the races examined in the three GSE datasets? There should be some racial differences in gene expression, which could modify the clinico-pathological expressions of MS among races. 

Author Response

Response to comments from Reviewer #2
We would also like to thank Reviewer #2 for his/her insights on the manuscript. We appreciate the
time and efforts put into reviewing our manuscript and value comments raised by this reviewer. Pointto-point replies to the comments/criticisms raised are shown below.
The authors analyzed the combined data from three independent datasets (GSE38010, GSE32915
and GSE108000) by the Stouffer’s Z-score method, and found 1446 DEGs; 742 up-regulated and 704
down-regulated. The validation of the newly identified top up- and down-regulated DEGs by real-time
quantitative PCR revealed the MS subtype-specific differences in the expression of these DEGs.
Although the study results are interesting, I have the following comments.
1. The qPCR results showed that the expression of the target genes varied according to the
disease stage or subtype. This suggests that DEGs should be separately analyzed according to
the disease stage or subtype. For instance, SPMS-specific DEGs could be quite different from
those of RRMS.
RESPONSE: We completely agree with this reviewer’s comment, which also aligns with a similar
comment from Reviewer #1. As said (please see refer to the reply to comment #1 above) we were
unable to stratify DEGs based on MS subtype (as we did in validation experiments) due to the lack of
clinical data of the patients available from the deposited datasets used to perform our meta-analysis.
We also attempted to gather clinical information from published work by the authors who generated
these datasets, but most information was incomplete and/or not directly linked to each sample.
Nonetheless, our real-time qPCR (using defined MS subtypes) helped us unveiling how these
markers differ in the different subclinical MS entities, highlighting a diverse and heterogeneous
expression pattern. Unfortunately, this is a limitation that is beyond our control but that certainly
pinpoints the need of more studies addressing this scientific need.
2. Why did the KEGG pathway analysis using DES in the combined GSE datasets show no
statistically significant associations with any specific intracellular pathway?
RESPONSE: This is a legitimate issue that requires explanations. Fundamentally, there are two main
reasons why the KEGG analyses of combined datasets were not statistically significant: (1) Biological
and (2) Bioinformatics/technical.
Biological explanation: It is already difficult to attain low inter-sample variability when analyzing data
from post-mortem WM samples due to the natural heterogeneity of the disease, the different postmortem intervals, the duration of disease etc. In this study, we collected datasets from different
sources, different collection methods and likely storage conditions, who may have contributed to
further increase the heterogeneity of results and hence, negatively impacted the statistical power of
annotated genes for any specific intracellular pathway.
Bioinformatics/technical explanation: As the reviewer may be aware, KEGG database is organized to
contain several ‘lists’ of genes, whose expression is associated with the activation/suppression of
specific intracellular functions/pathways. Many of these genes are often represented only in a limited
number of pathways. Therefore, from a statistical point of view, any given intracellular pathway
requires either that several DEGs match that specific pathway (difficult to attain) and/or few genes
match that pathway but with very high statistical significance (more likely). This is in contrast with
Gene Ontology (GO) databases where genes are assigned to a large set of predefined bins
depending on their functional characteristics, making it easier to attain several annotations for any
given biological function than in KEGG.
Notwithstanding the lack of statistical significance (in the combined datasets only), the output still
shows some intracellular pathways likely to be disrupted, although without statistical support. Hence,
our decision to include this information in the manuscript.
Thanks for giving us the possibility to explain this point in more detail.
3. The average FCs of both the newly identified up- and down-regulated DEGs by the combined
GSE datasets were much smaller compared with the combined GSE datasets. Why did the
authors perform the GO and KEGG pathway analyses using only the newly identified DEGs?
What is the rationale for this? These DEGs occupied the only small part of the total DEGs as
determined by the combined GSE datasets.
RESPONSE: The broad scope of this study was to combined datasets obtained using limited number
of samples with the ultimate goal of identifying genes whose disruptions may have been overlooked
(due to small cohort size). As such, it was expected that the newly identified genes from the
combined dataset would not show huge FC differences, as these genes were below the statistical
threshold for significance when analysed in each individual dataset. With regards to GO and KEGG
analyses, these were carried out both using the DEGs from all the combined datasets (Fig. 2) and
using only the newly identified genes (Fig. 4). The reason to run enrichment analyses on “newly
identified DEGs” was to share information on the biological (GO) and intracellular pathways (KEGG)
that were found to be also disrupted in the MS WM but that could not be revealed using the individual
datasets.
4. As shown in Figure 3, the number of overlapped DEGs among each GSE dataset and the
combined dataset was very small. It suggests a large variation of DEGs among the studies
reflecting the disease heterogeneity, which makes the interpretation of the significance of the
discovered DEGs very difficult.
RESPONSE: We perfectly agree with Reviewer #2 on this point. As mentioned above, this is a
limiting aspect of our findings which, as the reviewer also mentions, heavily relies on the intrinsic
heterogeneity of the disease. However, the fact that we were still able to identify significant
disruptions of certain new regulatory genes and pathways using the merged datasets supports the
idea that, in spite of a biological (and likely technical) heterogeneity, some critical genes and
pathways that were previously overlooked have been unveiled, expanding the knowledge on the
transcriptional regulation of genes in the WM of people with MS. In addition, validation experiments
confirmed that, selected top up- or down-regulated genes from bioinformatics predictions were
effectively dysregulated, hence strengthening the predictive value of the meta-analysis.
5. The qPCR data suggest that the newly discovered up-regulated DEGs are mostly related to
SPMS. The authors should describe the relevant pathological changes in SPMS corresponding
to the up-regulation of these DEGs.
RESPONSE: As Reviewer #2 noticed, validation of newly identified up-regulated genes (but not
down-regulated genes) were in part contributed by results from SPMS cases (2 out of 4 genes,
EPHX4 and L1CAM; Fig. 5B’ and 5C’). We are not quite sure if the reviewer overlooked this specific
point, as this was already discussed in the appropriate section. Please note that discussions on
EPHX4 up-regulation were more kept to a more general level (not specific to SPMS cases) as EPHX4
up-regulation (associated with improved lipid metabolism), was counteracted by a robust downregulation of VLDLR (also linked to lipid metabolism), therefore the conclusions we could make were
very limited (and speculative) in relation to this gene expression change if we attempted to link it to a
specific MS subtype (SPMS). To clarity these points to the reviewer, excerpts from the discussion
related to this point have been included below:
Page 12, lines 328-335: “…Two genes associated with cholesterol/lipid metabolism appeared
amongst the top 15 of the 175 newly detected DEGs, namely EPHX4 and VLDLR, with EPHX4 being
upregulated in SPMS cases and the expression for VLDLR being downregulated in all subtypes of MS
[26,34]. Impaired lipid metabolism has been linked to disease progression in MS, and was found to
have strong predictive value in determining the degree of neuro-degeneration and lesions’ formation
[52-56]. Cholesterol is an essential lipid for the synthesis of myelin and aberrant cholesterol
metabolism is linked to both impaired remyelination and myelin-debris phagocytosis [37,57-60].”
Page 12, lines 350-359: “…L1CAM is a gene that encodes for a cell-adhesion molecule that we found
to be significantly upregulated in the WM of SPMS patients only. Although it is not essential for ongoing myelination, L1CAM plays an important role in the initiation of myelination and supports the
interactions between axons and oligodendrocytes, a process that appears to be activated upon the
proteolysis of L1CAM protein [31,66,67]. Since L1CAM proteolytic products (L1 molecules) have been
detected in brain lysates, the increased expression of L1CAM in SPMS might suggest the increased
need for cleaved L1CAM proteins to restore homeostasis in patients with this progressive form of MS
[68].
6. What are the races examined in the three GSE datasets? There should be some racial differences
in gene expression, which could modify the clinico-pathological expressions of MS among races.
RESPONSE: Unfortunately, these information were not made available in the publicly available
datasets, nor specific information about the ethnicity of donors was found in any of the published
works using these datasets. It is agreeable that such additional data would have provided racespecific variations in gene expression profiles across different races. Hopefully, with the advent of
multiomics approaches, future publicly available datasets will offer more comprehensive details on
donors, allowing better stratification of data. We did our best to capture such information from the
samples used for validation, which were added in the new Table 3 in the revised manuscript.
We hope that the reviewers will be acknowledge our efforts to address the issues raised during the
evaluation of this study and will positively receive the revised submission.
With kind regards,
A/Prof. Alessandro Castorina
(corresponding author)
